# Layer-Parallel Training of Residual Networks with Auxiliary Variables

**Qi Sun**[1,2], **Hexin Dong**[2], **Zewei Chen**[3], **Weizhen Dian**[2], **Jiacheng Sun**[3], **Yitong Sun**[3],
**Zhenguo Li**[3] **and Bin Dong**[2]
[1]Tongji University, [2]Peking University, [3]Huawei Noah's Ark Lab
qsun_irl@tongji.edu.cn

## Abstract

Backpropagation algorithm is indispensable for training modern residual networks (ResNets) and usually tends to be time-consuming due to its inherent algorithmic lockings. Auxiliary-variable methods, *e.g.*, the penalty and augmented Lagrangian (AL) methods, have attracted much interest lately due to their ability to exploit layer-wise parallelism. However, we find that large communication overhead and lacking data augmentation are two key challenges of these approaches, which may lead to low speedup and accuracy drop. Inspired by the continuous-time formulation of ResNets, we propose a novel serial-parallel hybrid (SPH) training strategy to enable the use of data augmentation during training, together with downsampling (DS) filters to reduce the communication cost. This strategy first trains the network by solving a succession of independent sub-problems in parallel and then improve the trained network through a full serial forward-backward propagation of data. We validate our methods on modern ResNets across benchmark datasets, achieving speedup over the backpropagation while maintaining comparable accuracy.

## 1 Introduction

Gradient-based algorithms for training ResNets typically require a forward pass of the input data, followed by back-propagating [41] the gradient to update model parameters, which are often time-consuming as the depth goes further deeper. As such, many parallelization techniques including, but not limited to data-parallelism [20], model-parallelism [7], and a combination of both [37, 13, 35, 17] have been proposed to accelerate the training. Unfortunately, none of them could tackle the scalability barrier created by the intrinsically serial propagation of data within the network [9], preventing us from updating layers in parallel and fully leveraging the distributed computing resources.

Note that the backward pass takes roughly twice as long as the forward one and thus recent works have been focused on breaking the backward locking. One way is to apply the synthetic gradients to build decoupled neural interfaces [21], however, it fails in training deep convolutional neural networks (CNNs) [33]. [19] employs the delayed gradients for parameter updates but suffers from large memory consumption and weight staleness [27]. [18, 45] compensate the gradient delay by replaying the forward pass, which would incur additional computation cost. Another related work is the local error learning methods [34, 1, 2, 40, 29], but the objective loss function is not consistent with the original one and the forward dependency of a particular input data still exists.

Due to their ability to exploit layer-wise parallelism in both the forward and backward modes, the auxiliary-variable methods [4, 46, 6, 43, 47, 32, 25] have attracted much interest lately but is found to suffer from accuracy drop when training deep CNNs [11]. In other recent works [9, 38, 23], based on the similarity of ResNets training to the optimal control of nonlinear systems [8], parareal method for solving differential equations is employed to replace the forward and backward passes with iterative multigrid schemes. Since the feature maps need to be recorded and then used in a subsequent process

Presented at *Deep Learning and Differential Equations Workshop*, NeurIPS 2021.

$$\boxed{\arg\min\,\{\text{classification loss} \,|\, \text{ResNet} + \text{input images}\}}$$

discrete to $\downarrow$ continuum

$$\boxed{\arg\min\,\{\text{terminal loss} \,|\, \text{neural ODE} + \text{initial values}\}}$$

parallel-in-time $\downarrow$ or parareal methods

$$\boxed{\arg\min\,\{\text{terminal loss} + \text{intermediate penalties} \,|\, \text{local neural ODEs} + \text{auxiliary variables}\}}$$

continuum $\downarrow$ to discrete

$$\boxed{\arg\min\,\{\text{classification loss} + \text{stage-wise synthetic losses} \,|\, \text{stacked layers} + \text{auxiliary variables}\}}$$

Figure 1: A dynamical systems view of the construction of layer-parallel training algorithms.

to solve the adjoint equations, experiments were conducted on simple ResNets across small datasets. So far, to the best of our knowledge, it is uncertain that whether these auxiliary-variable methods can be effectively and efficiently applied to modern deep networks across real-world datasets.

In this work, we observe that there are two key issues that prevent us from attaining good practical performance via auxiliary-variable methods. The accuracy drop of trained model is mainly due to the lack of data augmentation, which is hard to implement at the presence of auxiliary variables. Moreover, data communication is another potential issue that may hamper the speed-up ratio, which was not adequately addressed in previous studies since most implementations were conducted only on CPUs. Based on these observations and inspired by the continuous-time reformulation of ResNets, we propose a novel DS-SPH training strategy that alternates between the traditional layer-serial training with data augmentation and the layer-parallel training in a reduced parameter space. Experimental results are carried out to demonstrate the effectiveness and efficiency of our proposed methods.

## 2 Method

**Preliminaries** Based on the concept of modified equations [8] or the variational analysis using $\Gamma$-convergence [44], training of ResNets from the scratch [14], *i.e.*,

$$\arg\min_{\{W_\ell\}_{\ell=0}^{L-1}} \left\{ \varphi(X_L) \,\middle|\, X_0 = S(y),\; X_{\ell+1} = X_\ell + F(X_\ell, W_\ell)\; \text{for } 0 \le \ell \le L \right\} \tag{1}$$

can be regarded as the discretization of a terminal control problem governed by the neural ODE [5]

$$\arg\min_{\omega_t} \left\{ \varphi(x_1) \,\middle|\, x_0 = S(y),\; dx_t = f(x_t, w_t)dt \;\text{on}\; (0,1] \right\} \tag{2}$$

where $\varphi(\cdot)$ denotes the classification loss function (see Appendix A for notation description). Furthermore, the continuous-time counterpart of the most commonly used backpropagation algorithm [16] for solving (1) is handled by the adjoint and control update equations for finding the extremal of problem (2) [26] (see Appendix A for more details).

As a direct result, the *locking effects* (*i.e.*, the forward, backward, and update lockings) [21] inherited from the serial propagation of data across all the building layers can be recast as the necessity of solving both the forward-in-time neural ODE (2) and a backward-in-time adjoint equation in order to perform the control updates (see Appendix A). Therefore, parallelizing the iterative system for solving the continuous-time optimization problem (2), *e.g.*, the parallel-in-time or parareal methods [31, 3], is a promising approach to achieve forward, backward, and update unlocking (see Figure 1).

To employ $K \in \mathbb{N}_+$ independent processors for the evolution of neural ODE in (2), we introduce a partition of $[0,1]$ into several disjoint intervals (see Figure 2), *i.e.*, $0 = s_0 < \ldots < s_k < \ldots < s_K = 1$, and define piecewise states $\{x_t^k\}_{k=0}^{K-1}$ such that the underlying dynamic evolves according to [1]

$$x_{s_k^+}^k = \lambda_k, \qquad dx_t^k = f(x_t^k, w_t^k)dt \;\text{on}\; (s_k, s_{k+1}], \tag{3}$$

*i.e.*, the continuous-time forward pass that originates from auxiliary variable $\lambda_k$ and with control variable $w_t^k$. This immediately implies that the optimization problem (2) can be reformulated as

$$\arg\min_{\{w_t^k\}_{k=0}^{K-1}} \left\{ \varphi(x_{s_K^-}^{K-1}) \,\middle|\, x_{s_k^-}^{k-1} = \lambda_k \;\text{and}\; x_{s_k^+}^k = \lambda_k,\; dx_t^k = f(x_t^k, w_t^k)dt \;\text{on}\; (s_k, s_{k+1}] \right\} \tag{4}$$

---

[1] Here, $x_{s_k^+}^k$ and $x_{s_k^-}^k$ refer to the right and left limits of the possibly discontinuous function $x_t^k$ at $t = s_k$.

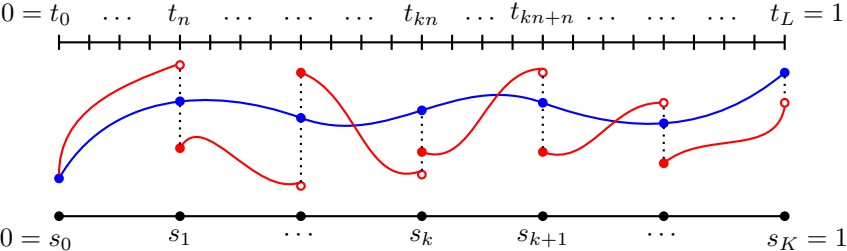

Figure 2: Contrary to the trajectory of neural ODE (blue line), introducing auxiliary variables (solid red dots) for each sub-interval (red lines) enables a time-parallel computation of the state, adjoint, and control variables. Note that to approximately recover the original dynamic, violation of equality constraints (mismatch between solid and hollow red dots) should be penalized in the loss function.

where $x_{s_0^-}^{-1} = \lambda_0 = x_0$. Such an observation offers the possibility of parallelizing the evolution of dynamical system (3) by relaxing the other constraint, *e.g.*, Figure 2 with external auxiliary variables.

**Layer-Parallel Training** The discussion above inspires us to loosen the exact connection between adjacent sub-intervals and add penalties to the objective function, that is,

$$\underset{\{w_t^k, \lambda_k\}_{k=0}^{K-1}}{\arg\min} \left\{ \varphi(x_{s_K^-}^{K-1}) + \beta \sum_{k=0}^{K-1} \psi(\lambda_k, x_{s_k^-}^{k-1}) \,\Big|\, x_{s_k^+}^k = \lambda_k, \; dx_t^k = f(x_t^k, w_t^k)dt \; \text{on} \; (s_k, s_{k+1}] \right\} \quad (5)$$

where $\beta > 0$ is a scalar constant and $\psi(\lambda, x) = \|\lambda - x\|_{\ell_2}^2$ the quadratic penalty function. Such a method has been extensively used due to its simplicity and intuitive appeal [43, 6, 11], however, it suffers from ill-conditioning when the penalty coefficient becomes large [36].

To make the approximate solution of (5) nearly satisfy the original problem (2) even for moderate values of coefficient $\beta$, we consider the augmented Lagrangian [36] of problem (4), namely,

$$\mathcal{L}_{AL} = \varphi(x_{s_K^-}^{K-1}) + \sum_{k=0}^{K-1} \left( \beta\psi(\lambda_k, x_{s_k^-}^{k-1}) + \int_{s_k}^{s_{k+1}} p_t^k \big(f(x_t^k, w_t^k) - \dot{x}_t^k\big)dt - \kappa_k(\lambda_k - x_{s_k^-}^{k-1}) \right)$$

where $p_t^k$ is the adjoint variable and $\kappa_k$ the explicit Lagrange multiplier. Notably, by forcing $\kappa_k \equiv 0$ for any $0 \le k \le K - 1$, the augmented Lagrangian method degenerates the penalty approach.

By calculus of variations [28], the iterative system for solving the relaxed problem (5) is provided in Appendix B, which leads to a non-intrusive layer-parallel training algorithm (see Appendix C) after employing the consistent discretization schemes [10] presented in Appendix A (see also Figure 1).

**Hybrid Training with Downsampling** We note that the introduction of explicit auxiliary variables can increase concurrency across all the building blocks, but it can also incur additional memory and communication overheads. This may limit the performance for the exposed parallelism especially in the setting of fine partitioned models with the use of complicated data augmentation techniques.

One way of reducing the data communication overhead is to design downsampling filters to attenuate the size of external auxiliary variables. To be specific, instead of transferring the full-size auxiliary variables between CPU and GPU cores, we can operate with the downsampled data

$$\Lambda_k = \text{DS}(\lambda_k), \; \text{or approximately,} \; \lambda_k \approx \text{US}(\Lambda_k)$$

to execute the forward pass (3), and hence the problem (5) can now be defined in a reduced space

$$\underset{\{w_t^k, \Lambda_k\}_{k=0}^{K-1}}{\arg\min} \left\{ \varphi(x_{s_K^-}^{K-1}) + \beta \sum_{k=0}^{K-1} \psi(\text{US}(\Lambda_k), x_{s_k^-}^{k-1}) \,\Big|\, x_{s_k^+}^k = \text{US}(\Lambda_k), \; dx_t^k = f(x_t^k, w_t^k)dt \; \text{on} \; (s_k, s_{k+1}] \right\}.$$

With a slight loss of accuracy, the memory and communication overheads are significantly reduced when compared with the original method (5). Moreover, the implementation is very straightforward and can be easily extended to the augmented Lagrangian method (omitted here for simplicity).

Another key observation is that each training sample requires to introduce a group of corresponding auxiliary variables. When cooperating with the commonly used data augmentation [42], it may incur prohibitive memory requirements. Unfortunately, most of the existing auxiliary variable methods fail to address this issue, which often leads to a significant accuracy drop of the trained networks [11].

To enable the use of data augmentation during training, we propose a serial-parallel hybrid training strategy that alternatives between the layer-serial and layer-parallel training modes as depicted in Figure 3. Although the serial portion hampers the speedup, it can compensate the constraint violations caused by downsampling and significantly increase the test accuracy.

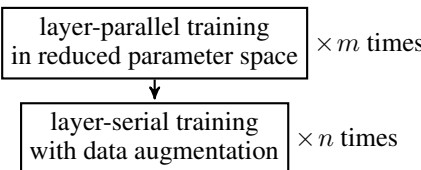

Figure 3: Hybrid training strategy.

## 3 Experiments

We conduct experiments using ResNet-110 on CIFAR-10 dataset [24] and report results to validate our methods (see Appendix D for more experimental results, *e.g.*, WideResNet on CIFAR-100 and image generation task). All our experiments are implemented in Pytorch 1.4 using multiprocessing library with NCCL backends [39]. The baseline network is split into $K$ pieces of stacked layers and then distributed on $K$ independent GPUs (Tesla-V100), which is trained for 200 epochs.

Table 1: Memory (GB), test accuracy, and speedup ratio (abbreviated as MEM, ACC, and SUR) of the proposed methods on ResNet-110 (the baseline model with ACC $= 93.7\%$) across CIFAR-10.

| Method | | MEM | ACC | SUR | | Method | | MEM | ACC | SUR |
|---|---|---|---|---|---|---|---|---|---|---|
| | K=2 | 1.53 | 85.0% | 1.41 | | | K=2 | 3.05 | 85.7% | 1.36 |
| P | K=3 | 4.58 | 84.5% | 1.56 | | AL | K=3 | 9.12 | 85.2% | 1.49 |
| | K=4 | 5.34 | 84.3% | 1.87 | | | K=4 | 10.68 | 83.7% | 1.64 |
| | K=2 | 0.38 | 84.2% | 1.53 | | | K=2 | 0.57 | 84.1% | 1.42 |
| DS-P | K=3 | 1.44 | 83.7% | 1.69 | | DS-AL | K=3 | 2.29 | 83.7% | 1.58 |
| | K=4 | 1.64 | 80.3% | 2.01 | | | K=4 | 2.67 | 80.0% | 1.74 |
| | K=2 | 1.53 | 91.8% | 1.30 | | | K=2 | 3.05 | 91.8% | 1.27 |
| SPH-P | K=3 | 4.58 | 91.3% | 1.40 | | SPH-AL | K=3 | 9.12 | 91.4% | 1.35 |
| | K=4 | 5.34 | 91.6% | 1.59 | | | K=4 | 10.68 | 91.2% | 1.45 |
| | K=2 | 0.38 | 91.8% | 1.38 | | | K=2 | 0.57 | 91.6% | 1.31 |
| DS-SPH-P | K=3 | 1.44 | 91.8% | 1.48 | | DS-SPH-AL | K=3 | 2.29 | 91.5% | 1.41 |
| | K=4 | 1.64 | 91.5% | 1.67 | | | K=4 | 2.67 | 91.0% | 1.51 |

As can be seen from Table 1, a straightforward implementation of the penalty (abbreviated as P) and AL methods requires a large memory capacity for storing the auxiliary variables and suffers from accuracy drop, which can be attenuated through the use of DS and SPH ($m = 4n$) methods. To further improve the accuracy, one can increase the serial portion but at the cost of slowing down the training process (see Table 2).

Table 2: ACC and SUR of ResNet-110 on CIFAR-10.

| | | $m : n$ (in the case of K=4) | | | |
|---|---|---|---|---|---|
| | | 4 : 1 | 2 : 3 | 3 : 2 | 1 : 4 |
| DS-SPH-P | ACC | 91.5% | 92.2% | 93.1% | 93.6% |
| | SUR | 1.67 | 1.43 | 1.25 | 1.11 |
| DS-SPH-AL | ACC | 91.0% | 92.0% | 92.8% | 93.6% |
| | SUR | 1.51 | 1.34 | 1.20 | 1.09 |

## 4 Conclusion

In this paper, we observed that the key issues that hampered the practicality of auxiliary-variable methods were data augmentation and communication. We then proposed a novel hybrid training strategy combined with downsampling to resolve these issues, and demonstrated the effectiveness of the proposed method on training large ResNets on CIFAR datasets. Potential future directions include investigation on the proposed method with more heavy duty deep ResNets, larger number of stages, exploring other choices of downsampling operators, other layer-parallel training algorithms, etc.

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

# Supplementary Materials

## A  Layer-Serial Training of Residual Networks

Without loss of generality, we consider the benchmark residual learning framework [14, 15] that assigns pixels in the raw input image to categories of interest as depicted in Figure 4. Its continuous-time analogue [44, 8] is then introduced to bridge such an image classification task with a terminal control problem constrained by the so-called neural ordinary differential equation [5].

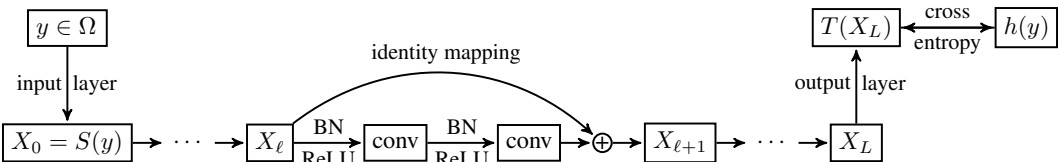

Figure 4: A diagram describing the serial training process of a pre-activation ResNet.

Given a human-labeled database $\{y, h(y)\}_{y \in \Omega}$, the optimization of model parameters requires solving (1), *i.e.*,

$$\underset{\{W_\ell\}_{\ell=0}^{L-1}}{\arg\min} \left\{ \mathbb{E}_{y \in \Omega} \Big[ \|T(X_L) - h(y)\| \Big] \,\Big|\, X_0 = S(y), \ X_{\ell+1} = X_\ell + F(X_\ell, W_\ell) \ \text{for } 0 \leq \ell \leq L-1 \right\}$$

where $X_\ell$ indicates the input feature map of the $\ell$-th building module, $L \in \mathbb{N}_+$ the total number of modules, $F$ typically a composition of linear and nonlinear functions as depicted in Figure 4, $W_\ell$ the network parameters to be learned, and $\|\cdot\|$ a given metric measuring the discrepancy between the model prediction $T(X_L)$ and the ground-truth label $h(y)$ for each training sample $y \in \Omega$. The trainable parameters of input and output layers, *i.e.*, $S$ and $T$ in Figure 4, are assumed to be fixed [12] for the ease of illustration.

When the most commonly used backpropagation algorithm [16] is applied to solving the optimization problem (1), we obtain the formula for parameter updates, *i.e.*,

$$W_\ell \leftarrow W_\ell - \eta \frac{\partial \varphi(X_L)}{\partial W_\ell} = W_\ell - \eta \frac{\partial \varphi(X_L)}{\partial X_{\ell+1}} \frac{\partial X_{\ell+1}}{\partial W_\ell}, \qquad 0 \leq \ell \leq L-1, \tag{6}$$

where $\eta > 0$ is the learning rate and $\varphi(X_L) = \mathbb{E}_{y \in \Omega} \Big[ \|T(X_L) - h(y)\| \Big]^2$.

Note that by defining $P_{\ell+1} = \dfrac{\partial \varphi(X_L)}{\partial X_{\ell+1}}$ for $0 \leq \ell \leq L-1$, formula (6) can be rewritten as

$$W_\ell \leftarrow W_\ell - \eta \left( P_{\ell+1} \frac{\partial F(X_\ell, W_\ell)}{\partial W} \right), \qquad 0 \leq \ell \leq L-1, \tag{7}$$

where $\{P_{\ell+1}\}_{\ell=0}^{L-1}$ satisfy a backward dynamic that captures the objective changes with respect to the hidden neurons, *i.e.*,

$$P_\ell = P_{\ell+1} \frac{\partial X_{\ell+1}}{\partial X_\ell} = P_{\ell+1} + P_{\ell+1} \frac{\partial F(X_\ell, W_\ell)}{\partial X}, \qquad P_L = \frac{\partial \varphi(X_L)}{\partial X_L}. \tag{8}$$

To put it differently, the full layer-serial backpropagation algorithm (6) is handled by formulae (8) and (7). As such, the training of ResNets at each iteration step requires the repeated execution of

- forward pass in (1)   • backward gradient propagation (8)   • parameter updates (7)

which can be very time-consuming as it is common to see neural networks with hundreds or even thousands of layers.

### A.1  Optimal Control of Neural Ordinary Differential Equations

The continuous-time counterpart of the minimization problem (1) is formulated as (2), that is,

$$\underset{\omega_t}{\arg\min} \left\{ \mathbb{E}_{y \in \Omega} \Big[ \|T(x_1) - h(y)\| \Big] \,\Big|\, x_0 = S(y), \ dx_t = f(x_t, w_t)dt \ \text{for } 0 < t \leq 1 \right\}$$

where the forward propagation through the underlying network with fixed parameters, *i.e.*, the constraint of (1) is interpreted as a numerical discretization of differential equations [8, 30, 5].

---

[2]Though the population risk is of primary interest, we only have access to the empirical risk in practice. For notational simplicity, we still denote by $\varphi(\cdot)$ the objective function obtained from a mini-batch of the entire training data throughout this work.

By introducing the Lagrange functional with multiplier $p_t$ [36], solving the constrained optimization problem (2) is equivalent to finding saddle points of the following Lagrange functional without constraints[3]

$$\mathcal{L}(x_t, w_t, p_t) = \varphi(x_1) + \int_0^1 p_t \left( f(x_t, w_t) - \dot{x}_t \right) dt$$

$$= \varphi(x_1) - p_1 x_1 + p_0 x_0 + \int_0^1 p_t f(x_t, w_t) + \dot{p}_t x_t \, dt.$$

and the variation in $\mathcal{L}(x_t, w_t, p_t)$ corresponding to a variation $\delta w$ in control $w$ takes on the form [28]

$$\delta \mathcal{L} = \left[ \frac{\partial \varphi(x_1)}{\partial x} - p_1 \right] \delta x + \int_0^1 \left( p_t \frac{\partial f(x_t, w_t)}{\partial x} + \dot{p}_t \right) \delta x + \left( p_t \frac{\partial f(x_t, w_t)}{\partial w} \right) \delta w \, dt,$$

which leads to the necessary conditions for $w_t = w_t^*$ to be the extremal of $\mathcal{L}(x_t, w_t, p_t)$, i.e.,

$$dx_t^* = f(x_t^*, w_t^*)dt, \qquad\qquad x_0^* = S(y), \qquad\qquad \text{(state equation)}$$

$$dp_t^* = -p_t^* \frac{\partial f(x_t^*, w_t^*)}{\partial x} dt, \qquad\qquad p_1^* = \frac{\partial \varphi(x_1^*)}{\partial x}, \qquad\qquad \text{(adjoint equation)}$$

$$p_t^* \frac{\partial f(x_t^*, w_t^*)}{\partial w} = 0, \qquad\qquad 0 \le t \le 1. \qquad\qquad \text{(optimality condition)}$$

However, directly solving this optimality system is computationally infeasible due to the so-called curse of dimensionality, a gradient-based iterative approach with step size $\eta > 0$ is typically used, e.g.,

$$dx_t = f(x_t, w_t)dt, \qquad\qquad x_0 = S(y), \qquad \text{(forward pass)} \qquad\qquad \text{(10a)}$$

$$dp_t = -p_t \frac{\partial f(x_t, w_t)}{\partial x} dt, \qquad\qquad p_1 = \frac{\partial \varphi(x_1)}{\partial x}, \qquad \text{(backward gradient propagation)} \qquad \text{(10b)}$$

$$w_t \leftarrow w_t - \eta \left( p_t \frac{\partial f(x_t, w_t)}{\partial w} \right), \qquad 0 \le t \le 1, \qquad \text{(parameter updates)} \qquad\qquad \text{(10c)}$$

which is consistent with the layer-serial training of ResNet through forward-backward propagation, i.e., (1), (8) and (7), by taking the limit as $L \to \infty$ [26]. In other words, the backprapagation method (6) can be recovered from (10b) and (10c) by employing the stable discretization schemes (8) and (7).

As a result, the *locking* effects [21] for training feedforward neural networks, i.e.,

(i) forward locking: no module can process its incoming data before the previous node in the directed forward network have executed;

(ii) backward locking: no module can capture the objective changes with respect to its activation layer before the previous node in the backward network have executed;

(iii) update locking: no building modules can be updated before all the dependent nodes have executed in both the forward and backward modes;

can be recast as the necessity of solving both the neural ODE in (2) and a backward-in-time adjoint equation (10b) in order to perform the control updates (10c).

This connection not only brings us a dynamical system view of the locking effects but also provides a way to consistently discretize the iterative system (10b) and (10c) for solving the continuous-time optimization problem (2). Therefore, breaking the locking issues, or, equivalently, parallelizing the iterative system for solving (2) is a promising approach to speed up the network training.

## B  Augmented Lagrangian Method

Recall that the neural ODE-constrained optimization problem (2) can be reformulated as (4), that is,

$$\arg\min_{\{w_t^k\}_{k=0}^{K-1}} \left\{ \varphi(x_{s_K^-}^{K-1}) \,\middle|\, x_{s_k^-}^{k-1} = \lambda_k \ \text{and} \ x_{s_k^+}^k = \lambda_k, \ dx_t^k = f(x_t^k, w_t^k)dt \ \text{on} \ (s_k, s_{k+1}] \ \text{for} \ 0 \le k \le K-1 \right\},$$

whose the augmented Lagrangian functional is expressed as

$$\mathcal{L}_{\mathrm{AL}}(x_t^k, p_t^k, w_t^k, \lambda_k, \kappa_k) = \varphi(x_{s_K^-}^{K-1}) + \sum_{k=0}^{K-1} \left( \beta \psi(\lambda_k, x_{s_k^-}^{k-1}) - \kappa_k(\lambda_k - x_{s_k^-}^{k-1}) + \int_{s_k}^{s_{k+1}} p_t^k \left( f(x_t^k, w_t^k) - \dot{x}_t^k \right) dt \right)$$

$$= \varphi(x_{s_K^-}^{K-1}) + \sum_{k=0}^{K-1} \left( \beta \psi(\lambda_k, x_{s_k^-}^{k-1}) - \kappa_k(\lambda_k - x_{s_k^-}^{k-1}) - p_{s_{k+1}^-}^k x_{s_{k+1}^-}^k + p_{s_k^+}^k \lambda_k + \int_{s_k}^{s_{k+1}} p_t^k f(x_t^k, w_t^k) + \dot{p}_t^k x_t^k \, dt \right).$$

---

[3]For notational simplicity, $\frac{dx_t}{dt}$ and $\dot{x}_t$ are used to denote the time derivative of $x_t$ throughout this work.

Specifically, the augmented Lagrangian functional can be decomposed as parts involving $x_t^{K-1}$ and $\{x_t^k\}_{k=0}^{K-2}$, i.e.,

$$I = \varphi(x_{s_K^-}^{K-1}) - p_{s_K^-}^{K-1} x_{s_K^-}^{K-1} + p_{s_{K-1}^+}^{K-1} \lambda_{K-1} + \int_{s_{K-1}}^{s_K} p_t^{K-1} f(x_t^{K-1}, w_t^{K-1}) + \dot{p}_t^{K-1} x_t^{K-1}\, dt,$$

and

$$II = \sum_{k=0}^{K-2} \left( \beta \psi(\lambda_{k+1}, x_{s_{k+1}^-}^k) + (\kappa_{k+1} - p_{s_{k+1}^-}^k) x_{s_{k+1}^-}^k + p_{s_k^+}^k \lambda_k + \int_{s_k}^{s_{k+1}} p_t^k f(x_t^k, w_t^k) + \dot{p}_t^k x_t^k\, dt - \kappa_{k+1} \lambda_{k+1} \right)$$

respectively, then the variation in $\mathcal{L}(x_t^k, p_t^k, w_t^k, \lambda_k, \kappa_k)$ corresponding to a variation $\delta w_t^k$ in control $w_t^k$ takes on the form

$$\delta \mathcal{L} = \left( \frac{\partial \varphi(x_{s_K^-}^{K-1})}{\partial x} - p_{s_K^-}^{K-1} \right) \delta x^{K-1} + \int_{s_{K-1}}^{s_K} \left( p_t^{K-1} \frac{\partial f(x_t^{K-1}, w_t^{K-1})}{\partial x} + \dot{p}_t^{K-1} \right) \delta x^{K-1}\, dt$$

$$+ \sum_{k=0}^{K-2} \left[ \left( \beta \frac{\partial \psi(\lambda_{k+1}, x_{s_{k+1}^-}^k)}{\partial x} + \kappa_{k+1} - p_{s_{k+1}^-}^k \right) \delta x^k + \int_{s_k}^{s_{k+1}} \left( p_t^k \frac{\partial f(x_t^k, w_t^k)}{\partial x} + \dot{p}_t^k \right) \delta x^k\, dt \right],$$

which implies that the adjoint variable $p_t^k$ satisfies the backward differential equations (11) [28], namely,

$$dp_t^k = -p_t^k \frac{\partial f(x_t^k, w_t^k)}{\partial x} dt \ \ \text{on } [s_k, s_{k+1}),$$

$$p_{s_{k+1}^-}^k = (1-\delta) \left( \beta \frac{\partial \psi(\lambda_{k+1}, x_{s_{k+1}^-}^k)}{\partial x} + \kappa_{k+1} \right) + \delta \frac{\partial \varphi(x_{s_{k+1}^-}^k)}{\partial x}, \tag{11}$$

for any $0 \leq k \leq K-1$. Here and in what follows $\delta = \delta_{k,K-1}$ represents the Kronecker Delta function.

Moreover, it can be easily deduced from the augmented Lagrangian functional that the control updates satisfy

$$w_t^k \leftarrow w_t^k - \eta \left( p_t^k \frac{\partial f(x_t^k, w_t^k)}{\partial w} \right) \quad \text{on } [s_k, s_{k+1}] \tag{12}$$

for $0 \leq k \leq K-1$, while the correction of auxiliary variables takes on the form

$$\lambda_0 \equiv x_0 \quad \text{and} \quad \lambda_k \leftarrow \lambda_k - \eta \left( \beta \frac{\partial \psi(\lambda_k, x_{s_k^-}^{k-1})}{\partial \lambda} + p_{s_k^+}^k - \kappa_k \right) \quad \text{for } 1 \leq k \leq K-1. \tag{13}$$

Notably, by choosing a quadratic penalty function $\psi(\lambda, x) = \|\lambda - x\|_{\ell_2}^2$, formula (13) shows that the constraint violations associated with the minimizer of augmented Lagrangian method satisfy for $1 \leq k \leq K-1$,

$$\lambda_k - x_{s_k^-}^{k-1} \approx \frac{1}{2\beta}(\kappa_k - p_{s_k^+}^k) \tag{14}$$

which offers two ways of improving the consistency constraint $x_{s_k^-}^{k-1} = x_{s_k^+}^k$: increasing $\beta$ or sending $\kappa_k \to p_{s_k^+}^k$, whereas the penalty method (by forcing $\kappa_k \equiv 0$ in (14), see also the formula (19) below) provides only one option. Moreover, it can be deduced from (14) that the update rule of explicit Lagrange multipliers satisfy

$$\kappa_0 \equiv 0 \quad \text{and} \quad \kappa_k \leftarrow \kappa_k - \frac{\eta}{2\beta}\left( \lambda_k - x_{s_k^-}^{k-1} \right) \quad \text{for } 1 \leq k \leq K-1. \tag{15}$$

In short, the augmented Lagrangian method for approximately solving problem (2) at each iteration step includes

- local operations (3), (11), (12) in parallel    • global communication (13), (15)

which not only parallelizes the iterative system (10) for solving (2) but also lessens the the issue of coefficient tuning.

## B.1  Penalty Method

Note that by forcing $\kappa_k \equiv 0$ for any $0 \leq k \leq K-1$, the augmented Lagrangian method degenerates to a penalty method. Specifically, it can be deduced from (11) that the adjoint equation for relaxed minimization problem (5) takes on the form

$$dp_t^k = -p_t^k \frac{\partial f(x_t^k, w_t^k)}{\partial x} dt \quad \text{on } [s_k, s_{k+1}),$$

$$p_{s_{k+1}^-}^k = (1-\delta)\beta \frac{\partial \psi(\lambda_{k+1}, x_{s_{k+1}^-}^k)}{\partial x} + \delta \frac{\partial \varphi(x_{s_{k+1}^-}^k)}{\partial x}, \tag{16}$$

for $0 \leq k \leq K - 1$. Moreover, by (12) and (13), the update rule for control variables now satisfies for $0 \leq k \leq K - 1$,

$$w_t^k \leftarrow w_t^k - \eta \left( p_t^k \frac{\partial f(x_t^k, w_t^k)}{\partial w} \right) \quad \text{on } [s_k, s_{k+1}], \tag{17}$$

while the correction of auxiliary variables is given by

$$\lambda_0 \equiv x_0 \qquad \text{and} \qquad \lambda_k \leftarrow \lambda_k - \eta \left( \beta \frac{\partial \psi(\lambda_k, x_{s_k^-}^{k-1})}{\partial \lambda} + p_{s_k^+}^k \right) \quad \text{for } 1 \leq k \leq K - 1. \tag{18}$$

In short, the penalty approach (5) for approximately solving problem (2) at each iteration consists of

> - local operations $(3), (16), (17)$ in parallel        • global communication (18)

which parallelizes the iterative system (10) for solving (2).

In particular, by choosing the quadratic penalty function $\psi(\lambda, x) = \|\lambda - x\|_{\ell_2}^2$ as before, it can be deduced from (18) that the constraint violations associated with the approximate minimizer of problem (5) satisfies for $1 \leq k \leq K - 1$,

$$\lambda_k - x_{s_k^-}^{k-1} \approx -\frac{1}{2\beta} p_{s_k^+}^k \tag{19}$$

which implies that a large penalty coefficient $\beta$ is needed in order to force the minimizer of (5) close to the feasible region of problem (2). By employing the augmented Lagrangian method (14), the ill-conditioning of penalty method can be lessened without increasing the penalty coefficient indefinitely, however, the introduction of external Lagrangian multipliers $\{\kappa_k\}_{k=0}^{K-1}$ requires additional memory and communication overheads that may hamper the speed-up ratio.

## C   Parallel Backpropagation and Communication

By utilizing the consistent finite difference schemes (see Appendix A) for the discretization of the time-parallel iterative systems established in Appendix B, we arrive at a layer-parallel training algorithm that enables us to fully leveraging the computing resources. The detailed derivations are presented in what follows.

Recall the partitioning of $[0, 1]$ associated with the original ResNet (1), *i.e.*,

$$0 = t_0 < t_1 < \ldots < t_\ell = \ell\Delta t < t_{\ell+1} < \ldots < t_{L=nK} = 1,$$

then the local sub-problem is built by choosing a coarsening factor $n > 1$ and extracting every $n$-th module as depicted in Figure 2, or, equivalently, the forward Euler discretization of neural ODE with the coarser gird

$$t_0 = s_0 < \ldots < s_k = t_{nk} < s_{k+1} < \ldots < s_K = t_L,$$

which can be implemented independently and trained with low accuracy at a correspondingly low cost[4].

To be specific, $[s_k, s_{k+1}]$ is uniformly divided into $n$ sub-intervals for $0 \leq k \leq K - 1$, *i.e.*,

$$s_k = t_{kn} < t_{kn+1} < \cdots < t_{kn+n-1} < t_{kn+n} = s_{k+1},$$

we have by (3) that feature flow of the $k$-th sub-network evolves according to

$$X_{kn}^k = \lambda_k, \qquad X_{kn+m+1}^k = X_{kn+m}^k + F(X_{kn+m}^k, W_{kn+m}^k) \tag{20}$$

where $0 \leq m \leq n - 1$. Then by using the particular numerical scheme (8) that arises from the discrete-to-continuum transition, the discretization of the adjoint equation (11) is given by a backward dynamic

$$P_{kn+m}^k = P_{kn+m+1}^k + P_{kn+m+1}^k \frac{\partial F(X_{kn+m}^k, W_{kn+m}^k)}{\partial X} = P_{kn+m+1}^k \frac{\partial X_{kn+m+1}^k}{\partial X_{kn+m}^k},$$

$$P_{kn+n}^k = (1-\delta) \left( \beta \frac{\partial \psi(\lambda_{k+1}, X_{kn+n}^k)}{\partial X} + \kappa_{k+1} \right) + \delta \frac{\partial \varphi(X_{kn+n}^k)}{\partial X}. \tag{21}$$

---

[4]The trainable parameters in the input and output layers, *i.e.*, $S$ and $T$, can be automatically updated by coupling into the first and last sub-problems respectively.

In other words, for any interval $[s_k, s_{k+1}]$ and arbitrary $0 \leq m \leq n$, the adjoint variable in (21) is equivalent to

$$P_{kn+m}^k = (1 - \delta)\left(\beta\frac{\partial\psi(\lambda_{k+1}, X_{kn+n}^k)}{\partial X_{kn+m}^k} + \kappa_{k+1}\frac{\partial X_{kn+n}^k}{\partial X_{kn+m}^k}\right) + \delta\frac{\partial\varphi(X_{kn+n}^k)}{\partial X_{kn+m}^k} \tag{22}$$

which captures the objective and layer-wise synthetic loss changes, namely, the second and the first term on the right-hand-side of (22), with respect to the latent states for $k = K - 1$ and $0 \leq k \leq K - 2$, respectively.

Contrary to the straightforward approach [31, 9] where the iterations are executed by first solving state equation (3), then adjoint equation (16) afterwards, and finally control updates (17), we conduct the control updates simultaneously with the solution of adjoint equation after solving the state equation.

Specifically, to discretize the update rule for control variables (12) for any $0 \leq k \leq K - 1$, i.e.,

$$w_t^k \leftarrow w_t^k - \eta\left(p_t^k\frac{\partial f(x_t^k, w_t^k)}{\partial w}\right) \quad \text{on } [s_k, s_{k+1}],$$

we adopt the numerical scheme (7) to guarantee the accurate gradient information [10], that is,

$$W_{kn+m}^k \leftarrow W_{kn+m}^k - \eta\left(P_{kn+m+1}^k\frac{\partial F(X_{kn+m}^k, W_{kn+m}^k)}{\partial W}\right)$$

$$= W_{kn+m}^k - \eta\left((1 - \delta)\left(\beta\frac{\partial\psi(\lambda_{k+1}, X_{kn+n}^k)}{\partial X_{kn+m+1}^k} + \kappa_{k+1}\frac{\partial X_{kn+n}^k}{\partial X_{kn+m+1}^k}\right) + \delta\frac{\partial\varphi(X_{kn+n}^k)}{\partial X_{kn+m+1}^k}\right)\frac{\partial X_{kn+m+1}^k}{\partial W_{kn+m}^k}$$

$$= W_{kn+m}^k - \eta\left((1 - \delta)\left(\beta\frac{\partial\psi(\lambda_{k+1}, X_{kn+n}^k)}{\partial W_{kn+m}^k} + \kappa_{k+1}\frac{\partial X_{kn+n}^k}{\partial W_{kn+m}^k}\right) + \delta\frac{\partial\varphi(X_{kn+n}^k)}{\partial W_{kn+m}^k}\right) \tag{23}$$

where the second equality holds by (20) and (22). Next, we have by (13) and (22) that the correction of auxiliary variables satisfies $\lambda_0 \equiv x_0$ and

$$\lambda_k \leftarrow \lambda_k - \eta\left(\beta\frac{\partial\psi(\lambda_k, X_{kn}^{k-1})}{\partial\lambda} + (1 - \delta)\left(\beta\frac{\partial\psi(\lambda_{k+1}, X_{kn+n}^k)}{\partial X_{kn}^k} + \kappa_{k+1}\frac{\partial X_{kn+n}^k}{\partial X_{kn}^k}\right) + \delta\frac{\partial\varphi(X_{kn+n}^k)}{\partial X_{kn}^k} - \kappa_k\right) \tag{24}$$

for $1 \leq k \leq K - 1$, while the update rule (15) for Lagrangian multiplier $\kappa_k$ is given by

$$\kappa_0 = 0 \quad \text{and} \quad \kappa_k \leftarrow \kappa_k - \frac{\eta}{2\beta}\left(\lambda_k - X_{kn}^{k-1}\right) \tag{25}$$

for $1 \leq k \leq K - 1$. Clearly, operations (24) and (25) require communication between adjacent layers which can impede the performance of parallel computations.

Consequently, the layer-parallel training approach for solving (1) can be formulated as the operations

> • local forward pass and backpropagation (20), (23)     • global communication (24), (25)

at each iteration, which breaks the forward, backward and update locking issues [21].

## C.1   Non-intrusive Implementation Details

The stage-wise parallel training approach is summarized in algorithm 1 for deep residual learning, which can be applied in a non-intrusive way w.r.t. the existing network architectures [5]. Besides, we denote by $K = 1$ the traditional method using the full serial forward-backward propagation.

**Downsampling for Data Communication** Note that for each iteration of algorithm 1, the computational time associated with the layer-serial ($K = 1$) and layer-parallel training methods can be summarized as follows:

|  | layer-serial | layer-parallel |
|---|---|---|
| forward pass | $t_f$ | $\frac{1}{K}t_f$ |
| backpropagation | $t_b$ | $\frac{1}{K}t_b + t_\psi$ |
| communication | $t_d$ | $t_\lambda + t_\kappa$ |

---

[5]To clarify the differences between layer-parallel training of fully-connected networks [47] and ResNets, we refer the readers to Figure 1 and algorithm 1 for technical details.

---

**Algorithm 1:** Layer-parallel training method for deep residual learning

```
// Initialization.
```
[1] divide the ResNet containing $L = Kn$ building modules into $K$ stages (see Figure 2);

[2] generate initial values for network parameters and auxiliary variables;

[3] set multipliers to zero; `// switch to penalty method if $\kappa_k \equiv 0$ hereafter`

[4] pick a suitable metric, *e.g.*, squared $\ell_2$-norm, $\ell_1$-norm or $\ell_\infty$-norm, for penalty function;

[5] choose positive sequences of increasing coefficients $\{\beta_j\}_{j=1}^J$ and decreasing tolerances $\{\tau_j\}_{j=1}^J$;

[6] schedule proper learning rates for network parameters, auxiliary variables and multipliers;

```
    // Training Procedure.
```
[7] **for** $j \leftarrow 1$ **to** $J$ *(number of epochs)* **do**

> ```
> // decoupled parallel forward-backward propagation on multiple GPUs
> ```
> **foreach** *mini-batch input data* **do**
>> **parfor** $k \leftarrow 0$ **to** $K - 1$ **do**
>>> ```
>>> // forward pass with auxiliary variable loading from CPU
>>> ```
>>> **for** $m \leftarrow 0$ **to** $n - 1$ **do**
>>>> $X_{kn}^k = \lambda_k^{\mathrm{mb}}, \quad X_{kn+m+1}^k = X_{kn+m}^k + F(X_{kn+m}^k, W_{kn+m})$
>>>
>>> ```
>>> // backpropagation with synthetic and objective loss functions
>>> ```
>>> **for** $m \leftarrow n$ **to** $0$ **do**
>>>> **if** $k \neq K - 1$ **then**
>>>>
>>>>> $W_{kn+m}^k \leftarrow W_{kn+m}^k - \eta \left( \beta_j \dfrac{\partial \psi(\lambda_{k+1}^{\mathrm{mb}}, X_{kn+n}^k)}{\partial W_{kn+m}^k} + \kappa_{k+1}^{\mathrm{mb}} \dfrac{\partial X_{kn+n}^k}{\partial W_{kn+m}^k} \right)$
>>>>
>>>> **else**
>>>>
>>>>> $W_{kn+m}^k \leftarrow W_{kn+m}^k - \eta \dfrac{\partial \varphi_{\mathrm{mb}}(X_{kn+n}^k)}{\partial W_{kn+m}^k}$
>
> ```
> // communicate across GPUs and data transmission from GPUs to CPU
> ```
> **for** $k \leftarrow 1$ **to** $K - 1$ **do**
>> **while** $\psi(\lambda_k^{\mathrm{mb}}, X_{kn}^{k-1}) > \tau_j$ **do**
>>> ```
>>> // update of auxiliary variables
>>> ```
>>> **if** $k \neq K - 1$ **then**
>>>
>>>> $\lambda_k^{\mathrm{mb}} \leftarrow \lambda_k^{\mathrm{mb}} - \eta \left( \beta_j \dfrac{\partial \psi(\lambda_k^{\mathrm{mb}}, X_{kn}^{k-1})}{\partial \lambda_k^{\mathrm{mb}}} + \beta_j \dfrac{\partial \psi(\lambda_{k+1}^{\mathrm{mb}}, X_{kn+n}^k)}{\partial X_{kn}^k} + \kappa_{k+1}^{\mathrm{mb}} \dfrac{\partial X_{kn+n}^k}{\partial X_{kn}^k} - \kappa_k^{\mathrm{mb}} \right)$
>>>
>>> **else**
>>>
>>>> $\lambda_k^{\mathrm{mb}} \leftarrow \lambda_k^{\mathrm{mb}} - \eta \left( \dfrac{\partial \varphi_{\mathrm{mb}}(X_{kn+n}^k)}{\partial X_{kn}^k} - \kappa_k^{\mathrm{mb}} \right)$
>>>
>>> ```
>>> // update of Lagrangian multipliers
>>> ```
>>> $\kappa_k^{\mathrm{mb}} \leftarrow \kappa_k^{\mathrm{mb}} - \dfrac{\eta}{2\beta_j} \left( \lambda_k^{\mathrm{mb}} - X_{kn}^{k-1} \right)$ `// for quadratic penalty function`

---

where $t_f$ ($t_b$) denotes the time cost of forward pass (backpropagation) using the layer-serial training method, $t_d$ the time cost on data loader, $t_\psi$ the computation time of synthetic loss functions, $t_\lambda + t_\kappa$ the evaluation and communication time of auxiliary variables. Clearly, the speedup ratio per epoch can be expressed as

$$\rho_K = \frac{\text{serial runtime}}{\text{parallel runtime}} = \frac{1}{\frac{1}{K} \frac{t_f + t_b}{t_f + t_b + t_d} + \frac{t_\psi + t_\lambda + t_\kappa}{t_f + t_b + t_d}} \tag{26}$$

where $t_f, t_b, t_d, t_\psi, t_\lambda$ and $t_\kappa$ are almost independent of the model partition number $K$ during training.

For realistic neural networks such as ResNets [14, 15], it is plausible to assume that $t_f + t_b + t_d > t_\psi + t_\lambda + t_\kappa$, which immediately shows speed-up over the traditional layer-serial training approach by choosing a sufficient large value of $K$. Notably, formula (26) also implies that the upper bound of speed-up ratio is given by

$$\rho_K < \frac{t_f + t_b + t_d}{t_\psi + t_\lambda + t_\kappa},$$

namely, the communication becomes the performance bottleneck as the model is partitioned more finely, which motivates us to reduce the data communication overhead in order to further accelerate the network training.

One way to achieve this is to design downsampling (DS) filters to attenuate the size of auxiliary variables. We can, for instance, take the example of penalty method (5). Instead of transferring the full-size auxiliary variables between CPU and GPU cores, we can operate with the downsampled data

$$\Lambda_k = \mathrm{DS}(\lambda_k), \ \text{ or approximately, } \ \lambda_k \approx \mathrm{US}(\Lambda_k)$$

to execute the forward pass (3) for $0 \leq k \leq K-1$

$$x^k_{s^+_k} = \mathrm{US}(\Lambda_k), \ \ dx^k_t = f(x^k_t, w^k_t)dt \ \text{ on } (s_k, s_{k+1}]. \tag{27}$$

For instance, by taking the Kronecker product with an all-ones matrix of size $2 \times 2$ for each slice of the tensor $\Lambda_k$, we obtain the auxiliary variable $\lambda_k$ for forward pass. section 3 will focus on this particular example and we leave the exploration of other downsampling tools as future work.

As such, the optimization problem is now defined in a reduced parameter space, that is,

$$\underset{\{w^k_t, \Lambda_k\}^{K-1}_{k=0}}{\arg\min} \left\{ \varphi(x^{K-1}_{s^-_K}) + \beta \sum^{K-1}_{k=0} \psi(\mathrm{US}(\Lambda_k), x^{k-1}_{s^-_k}) \Big| (27) \right\}.$$

With a slight loss of accuracy, the memory and communication overheads can be significantly reduced compared with the original method (5). Moreover, the implementation is very straightforward, only requiring an additional upsampling layer before the execution of forward pass in Algorithm 1, while the backpropagation is automatically achieved through the standard auto-differentiation. Such a technique can also be easily extended to the augmented Lagrangian method (omitted here for simplicity).

**Hybrid Training for Data Augmentation** To justify our argument about the lacking of data augmentation, we shown some preliminary experimental results by using different ratios $\rho_{\mathrm{DA}}$ of data augmentation (*i.e.*, the number of synthetic images to the number of real images), Figure 5 shows the test accuracy of ResNet-110 for the classification task on CIFAR-10 dataset, where $\rho_{\mathrm{DA}} = \infty$ denotes the data augmentation containing random operations. It can be observed that, as $\rho_{\mathrm{DA}}$ is increased, the accuracy gap between the traditional layer-serial training method and the proposed layer-parallel training approach is tending to close. However, the memory requirements for storing all the synthetic training data blows up even for moderate values of $\rho_{\mathrm{DA}}$, which is unaffordable in practical scenarios.

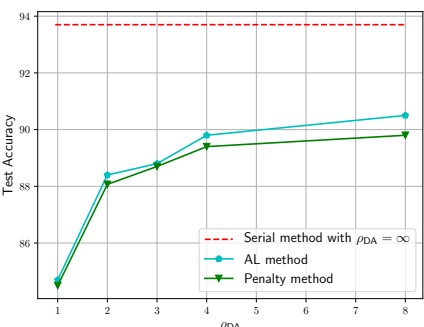

Figure 5: Test accuracy of trained model using data augmentation.

Unfortunately, most of the existing auxiliary-variable methods fail to address this issue, which often leads to a significant accuracy drop of the trained networks [11]. To allow the use of data augmentation during training, we propose a novel serial-parallel hybrid strategy that alternatives between the layer-serial and layer-parallel training modes as depicted in Figure 3. That is, the layer-parallel training method is performed $m$ times without data augmentation, followed by $n$ times execution of the layer-serial training method, which improves the network parameters through the employment of data augmentation.

As an immediate result, the speed-up ratio now gives

$$\rho_H = \frac{(m+n) \times t_s}{m \times t_p + n \times t_s} = \frac{1 + \gamma_H}{\frac{1}{\rho_K} + \gamma_H} \tag{28}$$

where $\gamma_H = \frac{n}{m}$ indicates the hybrid ration, $t_s = t_f + t_b + t_d$ and $t_p = \frac{1}{K}(t_s + t_b) + t_\psi + t_\lambda + t_\kappa$ are runtime of the layer-serial and layer-parallel training methods per epoch. Although the serial portions in (28) hamper the speedup ratio, *i.e.*, $\rho_H < \rho_K$, enabling data augmentation can significantly increase the test accuracy as shown in Figure 5. Moreover, the constraint violations caused by downsampling, *i.e.*, $\mathrm{US}(\Lambda_k)$ is applied to match $x^{k-1}_{s^-_k}$ in (7) instead of $\lambda_k$, can be adjusted through the layer-serial training procedure, which also works for applications without the use of data augmentation.

# D Additional Experiments

To further demonstrate the effectiveness of our proposed methods, we conduct experiments on various network architectures, training datasets, and learning tasks.

To begin with, we show in Figure 6 the learning curves of ResNet-110 on CIFAR-10 dataset, where $K = 3$ and $\gamma_H = 1/4$. As can be seen from Figure 6 (right) and Figure 7 (right), a large jump of constraint violation appears when the training is switched from parallel to serial, which helps improve the trained model through the use of data augmentation (see Figure 6 (left and middle)). Moreover, by using the same penalty coefficient, the constraint violation associated with AL method is smaller than that of penalty method (see Figure 6 (right)), which validates our theoretical analysis (19) and (14).

**Baseline** As a benchmark model for CIFAR-10 dataset, the network architecture of our baseline ResNet-110 is constructed as that described in [14], which achieves $93.7\%$ accuracy on the 10k testing images in 200 epochs. The total number of trainable parameters is around 1.7 million. Model is trained on the 50k training images with a batch size of 128, a weight decay of $10^{-4}$, momentum 0.9 and standard data augmentation. The training starts with a learning rate of $0.1$, and is divided by 10 at 50 and 150 epochs.

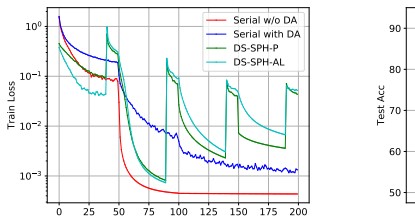 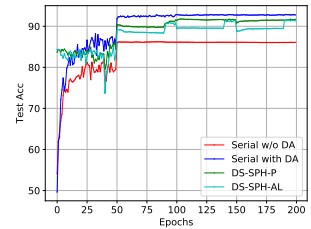 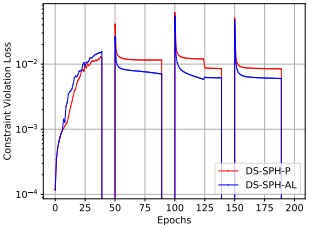

Figure 6: Training loss through the full serial forward pass, testing loss and constraint violation.

## D.1 Images Classification

Our methods also work well for WideResNet on CIFAR-100 dataset. The feature map of WideResNet-40-10 is roughly ten times larger than that of ResNet, which requires more auxiliary variables for training and makes the parallelization even more challenging. Table 3 implies that the performance gap between the serial training with data augmentation and vanilla parallel training, which roughly equals to 15%, is much larger than that of ResNet on CIFAR-10. Our methods can effectively decrease this gap to 4.48% and 3.7% for penalty and AL approaches, respectively. Furthermore, the speed-up ratio is larger than that of ResNet, say 1.76 compares to 1.48 for DS-SPH-P and 1.62 compares to 1.41 for DS-SPH-AL. This is because a large number of auxiliary variables naturally lead to larger memory cost and communication overhead, and our DS strategy can reduce much cost in them so as to exhibits a better speed-up ratio.

Table 3: MEM, ACC and SUR of WideResNet on CIFAR-100, where $K = 3$ and $\gamma_H = 1 : 4$.

| Method | MEM | ACC | SUR | Method | Memory | ACC | SUR |
|---|---|---|---|---|---|---|---|
| Serial w/o DA | - | 66.53 | - | Serial with DA | - | 80.71 | - |
| Penalty | 45.77 | 64.91 | 1.89 | AL | 91.55 | 64.80 | 1.67 |
| DS-P | 11.44 | 61.06 | 2.19 | DS-AL | 22.89 | 60.84 | 1.92 |
| SPH-P | 45.77 | 75.25 | 1.52 | SPH-AL | 91.55 | 76.25 | 1.42 |
| DS-SPH-P | 11.44 | 76.23 | 1.76 | DS-SPH-AL | 22.89 | 76.84 | 1.62 |

Learning curves are depicted in Figure 7, and the trade-off between speedup ratio and testing accuracy are shown in Table 4 and Figure 8.

## D.2 Image Generation

To demonstrate that our methods are not limited to image classification, we consider the image generation task in what follows. A VAE [22] model contains a pair of encoder and decoder. The serial model is implemented based on ResNet-VAE whose encoder is ResNet-110. The encoder for penalty method and AL method is divided to three stages and the initial penalty coefficient $\beta$ is set as 10. We use Adam as an optimizer for all networks with hybrid ratio $\gamma_H = 1:4$, and the initial learning rate is set as 0.01. To evaluate the generated image, we use the reconstructing MSE loss between reconstructed images and original images on the test set.

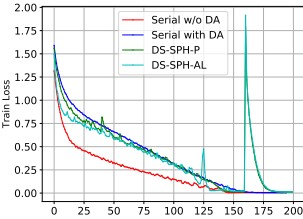 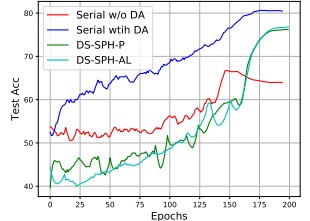 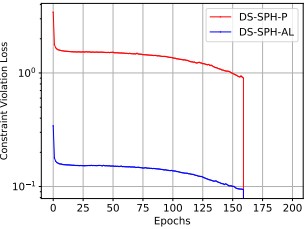

Figure 7: Training loss through the full serial forward pass, testing loss and constraint violation.

Table 4: ACC and SUR of WideResNet on CIFAR-100.

|  |  | $m : n$ (in the case of K=3) | | | |
|---|---|---|---|---|---|
|  |  | $4 : 1$ | $2 : 3$ | $3 : 2$ | $1 : 4$ |
| DS-SPH-P | ACC | 76.23% | 78.34% | 80.22% | 80.69% |
|  | SUR | 1.76 | 1.48 | 1.28 | 1.12 |
| DS-SPH-AL | ACC | 76.84% | 78.08% | 79.97% | 80.37% |
|  | SUR | 1.62 | 1.40 | 1.24 | 1.11 |

The results are shown in Table 5. As data augmentation is usually not used in the training of VAE, we omit the experiments with only SPH. As the results show, the vanilla penalty and AL, in general, can achieve acceptable performance on VAE, since data augmentation is not essential. However, they still suffer from enormous memory cost and CPU-GPU communication overhead. DS strategy reduces those costs significantly with a slight drop in performance. The reconstruction loss reduces from 0.149 to 0.160 for DS-P and from 0.129 to 0.137 for DS-AL. Combining with SPH, the performance drop caused by DS is almost eliminated. This result shows that SPH plays a more significant role than merely providing data augmentation. It also helps eliminate the gap between the real feature map and the auxiliary feature map and get better results, in Figure 9 we show the reconstructed images sampled from the test set to examine their visual qualities. From the figure, we see that our methods indeed reconstruct high-quality images.

Table 5: Test loss and SUR on MNIST generation.

| METHOD | TEST LOSS | SUR |
|---|---|---|
| BASELINE MODEL | 0.126 | - |
| PENALTY | 0.149 | 2.33 |
| AL | 0.129 | 2.18 |
| DS-P | 0.160 | 2.41 |
| DS-AL | 0.137 | 2.25 |
| DS-SPH-P | 0.127 | 1.88 |
| DS-SPH-AL | 0.128 | 1.80 |

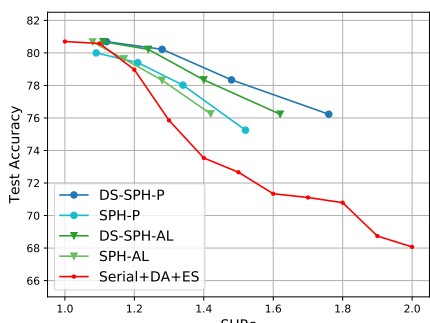

Figure 8: Accuracy-speed trade-off for different methods. The red-dot line is drawn by early stopping (ES) the serial training at epochs corresponding to the speed-up ratio.

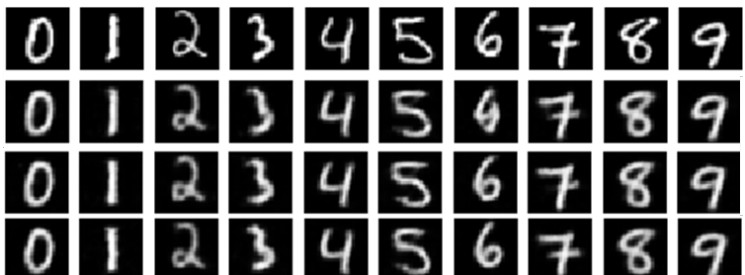

Figure 9: Reconstruction images in test set. **Line-1**: Original images; **Line-2**: Reconstruction from baseline model; **Line-3**: Reconstruction from DS-SPH-P; **Line-4**: Reconstruction from DS-SPH-AL.

