# OpenReview forum: "Layer-Parallel Training of Residual Networks with Auxiliary Variables"
_NeurIPS.cc/2021/Workshop/DLDE — DLDE Workshop -- NeurIPS 2021 Poster_

### Official Review · Reviewer_RiA8 · 2021-10-03

**Confidence:** 4

**Review:**

The authors propose and valiate a depth-parallel training scheme for ResNets, inspired by auxiliary variable numerical methods. The work further discusses limitations of existing penalty methods, and proposes a hybrid serial-parallel training strategy.

**Novelty & Significance**

Several layer-parallel schemes for ResNets have been proposed throughout the years [1,2], with limited practical success. As such, this is very much an open problem, though it is unclear whether this paper represents a step forward. The results are rather poor (Table 1, Figure 6). I commend the authors for the exposition of the method, which is clear.

**Questions**

* The hybrid strategy is necessary for accuracy (see Table 1). The authors should show performance (and memory footprint) of a baseline serial ResNet trained with same hyperparameters. Which baseline is the speedup ratio computed against?


**Minor**

The authors might find [3] interesting. [3] also proposes a parallel scheme not based on the penalty method, but for neural ODEs.

line 254: differerntial

[1] Parallel-in-time optimal control of neural networks, arXiv

[2] Layer-parallel training with GPU concurrency of deep residual neural networks via nonlinear multigrid, HPEC20

[3] Differentiable Multiple Shooting Layers, NeurIPS21

**Score:**

3: Good paper

---

### Official Review · Reviewer_sAab · 2021-10-11

**Confidence:** 2

**Review:**

The paper describes an approach to train ResNets efficiently, using auxiliary variables. The paper uses 2 methods -- penalty based and Augmented Lagrangian for parallel training of ResNet-110. To improve memory and time efficiency for utilizing data augmentation, downsampling of Auxiliary variables and hybrid Series-Parallel training are proposed.

**Comments**

The overall paper presentation is good and easy to understand. The idea to combining downsampling and hybrid serial-parallel training is interesting.


**Issues**

There are no details provided about ResNet-110 baseline training (against which the speedup ratio is computed).

 Some typos:

   Line 67: immediate -- immediately
   Line 108: staked -- stacked

**Score:**

3: Good paper

---

### Official Review · Reviewer_LmhB · 2021-10-11

**Confidence:** 4

**Review:**

In this work that authors propose a serial-parallel hybrid method to train Resnet, which utilizes the continuous time reformulation of Resnets. The model alternates between serial training (that enables data augmentation) and layer parallel training with auxiliary variable methods with downsampling the external auxiliary variables to reduce the communication overhead.
The authors show empirical results on ResNet 110 on the Cifar-10 dataset.

Overall I find the paper to be well written and motivated. Downsampling of the auxiliary variable is a good idea.

Few questions for the authors:
- I am not sure what the speed up ratio is calculated. Is it calculated against the baseline?
- How much are the auxiliary variables downsampled by?



**Score:**

3: Good paper

---

### Decision · Program_Chairs · 2021-10-14

**Decision:**

Accept (Poster)

**Comment:**

The reviews were generally positive, with several questions about the practicality of the proposed approach.